# Current State of Rare and Endangered Barbels of the Genus *Luciobarbus* Heckel, 1843 in the Aral–Syrdarya Basin (Kazakhstan) and Prospects for Their Conservation (A Review)

**DOI:** 10.3390/biology12040489

**Published:** 2023-03-23

**Authors:** Kamila Adyrbekova, Kuanysh B. Isbekov, Tynysbek T. Barakbayev, Galymzhan Zh. Iskhakhov

**Affiliations:** 1Department of Molecular Biology and Genetics, Faculty of Biology and Biotechnology, Al-Farabi Kazakh National University, Almaty 050040, Kazakhstan; 2Fisheries Research and Production Center LLP, Almaty 100141, Kazakhstan; 3Aral Branch of Fisheries Research and Production Center LLP, Kyzylorda 000090, Kazakhstan

**Keywords:** Kazakhstan, Aral Sea, Syrdarya River, barbels, rare species, endangered species, conservation, artificial reproduction, broodstock

## Abstract

**Simple Summary:**

In recent years, Kazakhstan has entered a period of rapid economic development, which will definitely lead to an increase in the anthropogenic pressure on natural resources. Subsequently, adequate measures must be taken to reduce the anthropogenic impact on natural resources, including the ichthyofauna. Therefore, the barbel was included in the Red Book of the Republic of Kazakhstan in 2008. Stocks of commercial fish valuable in the past such as barbels in the Aral–Syrdarya basin are now catastrophically reduced as a result of the anthropogenic impact on the ichthyofauna and natural environment of the Aral Sea region. Currently, the only way to restore the number of barbels is artificial reproduction, in order to preserve biodiversity and restore stocks of this valuable species in the natural environment and aquaculture.

**Abstract:**

The current status of the barbels inhabiting the Kazakhstan section of the Syrdarya River needs special study, as has been noted since the second half of the 20th century. Stocks of commercial fish valuable in the past as barbels in the Aral–Syrdarya basin are now catastrophically reduced as a result of anthropogenic impact on the ichthyofauna and natural environment of the Aral Sea region. The study of the condition; abundance and distribution range is necessary to determine measures for their restoration in natural habitats and their breeding in fish farms. Research on the biotechnology of barbel breeding with subsequent acclimatization and reacclimatization of these fish species will not only improve the composition of the ichthyofauna of the Aral–Syrdarya basin, but also preserve the genetic potential of natural populations. At present, the only way to restore the population of the Aral barbel is stocking hatchery reared juveniles in natural environments. Under the current situation, a way forward is seen in the creation of domesticated replacement-broodstocks of barbels. As a result of the influence of anthropogenic factors, the populations of this species have come to almost complete degradation and depletion and require the development and adoption of urgent measures for their conservation and restoration through reintroduction, which is a priority and urgent task for the fisheries of the republic.

## 1. Introduction

Two species of barbels of the genus *Luciobarbus* Heckel, 1843 of the family *Cyprinidae* (Cypriniformes) are found in water bodies in the territory of Kazakhstan: the Aral barbel—*Luciobarbus brachycephalus* Kessler, 1872 and the Bulatmai barbel *Luciobarbus capito* Güldenstädt, 1773, belonging to the Ponto-Caspian freshwater faunal complex and having largely coinciding distribution ranges [1,2,3]. The Aral barbel *L. brachycephalus* is represented by two populations. The first one distributed in the Caspian Sea basin and the second presented in the Aral basin. The Bulatmai barbel *L. capito* is also represented by two populations: in the southern part of the Caspian, but not in Kazakhstan and in the Aral Sea basin [4]. On the map (QGIS 3.22.13 V.), we consider only the Aral–Syrdarya basin (Figure 1).

These species are morphologically very close and differ only in a small number of characters. The basis for their separation is the ecology of habitat and lifestyle (migratory and residential forms). The systematic position, taxonomic rank of the barbels of the Aral–Syrdarya basin in several works of this group were carried out exclusively by the comparative morphological method; at the same time, the results of studies by different authors often differ significantly [5,6,7,8,9]. The variability of the morphological features of barbels of both species is very low, many characters overlap. An analysis of the morphometric traits of juveniles revealed that visually, the most significant traits are the number of scales in the lateral line [5]. According to the literature data, the issue of clarifying the taxonomy of the genus is still required [4] and, in our opinion, requires a molecular genetic analysis.

In previous times barbels were commercially important species of the Aral Sea and the Syrdarya River; nowadays they are listed in the Red Book of Kazakhstan. In this regard, studies of their current state are extremely limited, since special permission is required.

The Aral barbel, *L. brachycephalus* is sharply reduced in numbers and occupies the conservation status of category II and the fishing of this species is prohibited. Nowadays, the Bulatmai barbel is one of the species that is rapidly reducing in numbers in the republic [4]. On the IUCN list, the Aral barbel is listed as Critically Endangered, and the Bulatmai, in turn, as a vulnerable species [9].

It has been reported that the population of *L. brachycephalus* has declined by 30% over the past 30 years and continues to decline due to increased salinity of the Aral Sea and damming of its tributaries [10]. The Aral Sea was once the world’s fourth largest inland body of water in terms of surface area [11]. A lake basin, fed by two rivers, the Amu Darya and the Syrdarya, it supported a diverse ecosystem and an economically valuable fishery [12]. Nowadays the Aral Sea, is facing severe environmental degradation due to the rapid reduction in the water level. The main impact due to the water level receding is increasing salinity which reduces the biodiversity. Exposed lake beds become the source for fine dust picked up by the dust storms and are spread across a long distance [13].

The regulation of the flow of the Syrdarya and Amudarya rivers had a tremendous negative impact on the reproduction of the Aral population of barbel. Large water withdrawals for irrigation led to resorption of fish oocytes, and death of juveniles in irrigation systems. All of the above has led to a serious disruption of natural reproduction and a catastrophic reduction in the barbel’s population. Dams without fish passages blocked the path of the barbel to the traditional spawning grounds, and have reduced the natural recruitment to the population [14].

The natural range of the Aral barbel is the Aral basin, where, along with the typical anadromous type, there is form which lives constantly in rivers or lakes. In the sea, the barbel was found everywhere, especially in the coastal areas; it was found in the Syr Darya both in the river itself and its tributaries, and in the Shardara reservoir [4]. It should also be noted that in 1930–1931, the Aral barbel from the lower reaches of the Syrdarya was introduced into the Balkhash basin: more than 18 thousand underyearlings and two-year-olds were released into the Ile River near the present city of Konaev. In the first years after the introduction, the barbel was occasionally found in scientific fishing, was not reported in fishing until 1949, which suggested the failure of acclimatization. Later, barbel of different ages began to be found again in different areas [15,16,17,18]. In the early 1980s, the spawning schools in the lower reaches of the Ile River numbered about three thousand specimens [18]. In the Ile River above the Kapshagay reservoir in 1991–1993 the grown barbel was not observed, and its early juveniles were single in the samples [19].

The Bulatmai barbel inhabits the waters of the Aral basin, but unlike the Aral barbel, it is a typically freshwater, residential form that does not make distant migrations. It was rare in the Aral Sea, occurring mainly in coastal desalinated areas [4]. The Bulatmai barbel forms groups only during the spawning period, the rest of the time it lives alone or in small groups. It was noted in the flatland of the Syrdarya and Chu rivers, as well as in floodplain lakes, main lakes and in effluent channels of irrigation systems [20,21]. The Kazakh part of the range includes the Syrdarya basin from the Shardara reservoir to the lower reaches, including the basins of rivers flowing from the southwestern slopes of the Karatau ridge (Arys, Bogen) [4]. In the Arys River, expedition reports also noted the presence of these two species.

Prior to the regulation of the flow of the Syrdarya River, the main factors influencing the change in the abundance of the Aral barbel were the magnitude and nature of the river flow, which largely depended on the conditions for entry into the river of spawning fish and the migration of juveniles into the sea [22]. Irrigation construction on the rivers of the basin, which entailed the division of rivers into sections, a reduction in their flow, a drop in level, a decrease in the speed of the flow, and a decrease in turbidity, led to a serious disruption of the natural reproduction of the barbel. A huge number of juvenile barbels died in irrigation systems, since the slope coincided in time with the largest withdrawals of water from rivers for irrigation of fields. Constructions of the Kazalinsky dam have inundated the Syrdarya River and obstruct the reproductive migration of the Aral barbel [23,24,25,26,27,28,29,30,31].

According to archival materials, barbel was harvested in the Aral Sea all year round, but mainly (about 90% of the total fishing) during the period of mass spawning migration to river deltas. Its concentration for a rather long period (up to 3 months) and in relatively small areas created favorable opportunities for fishery.

The average fishing of barbel in the Aral Sea in previous decades was: 1140 tons in 1930–1939, but decreased to 326 tons in 1970–1979. Moreover, the catastrophic drop in the level of the Aral Sea and increasing salinity led to the fact that by the beginning of the 1980s, the Aral and Bulatmai barbels ceased to inhabit the sea itself [32].

The number of the Aral barbel in its natural habitat has been catastrophically reduced and is currently extremely low. The main factors that influenced the commercial stocks of barbel were: deterioration of the conditions of natural reproduction; illegal fishing; a large by-catch of juveniles in fixed nets; and mass mortality of juveniles in irrigation systems.

### Current State of Barbels in the Aral–Syrdarya Basin

According to 2003 data in the irrigation network of rice farms in the Karmakshinsky district of the Kyzylorda region, were more than 300 specimens of juvenile barbels, mainly underyearlings, 4–16 cm in size and weighing 3–45 g [32]. These juveniles were then released into the ponds of the Amanotkel fish farm to form a broodstock. At the same time, 43 individuals were subjected to biomorphological analysis, of which 25 were assigned to the Aral and 18 to the Bulatmai barbels [33].

There are places in the Syrdarya basin, it was noted, where barbels have significant numbers, for example the Bulatmai barbel was reported from irrigation systems of rice farms in the Kyzylorda region. When water is supplied to the irrigation system, mostly in May–July, barbels spread widely through the canals. Yearlings and older age groups, leaving the main canal, enter in large numbers into shallow waterways and rice fields. In addition, pelagic eggs and larvae of these species are passively brought into irrigation canals and fields. In August–September, the flow of water is reduced; the barbels move up the canals, trying to leave the irrigation systems in the Syrdarya [34]. However, the presence of hydraulic structures on the canals prevents the free exit of fish from the canals into the river. This migration becomes especially problematic when there is a sharp decrease in water inflow. According to our investigation in the field, a large number of juvenile barbels remain in the main and in the small irrigation canals, waste water bodies, and rice fields.

There was an expedition in August 2003 for collecting barbels from the drying irrigation systems of rice farms in the Karmashinsky district of the Kyzylorda region for fish farming. The group surveyed 11 points on the water supply and spillway irrigation network of rice farms, as well as a lake formed by drainage water from waste channels in the vicinity of Akzhar village. In the irrigation network, four specimens of the Aral and four specimens of the Bulatmai barbel were caught, ranging in size from 4 to 12 cm. fishing in August 2003 was hampered by high water levels and the late start of canal drainage. At several sites of the rice farms (“Bekenov and CO”, “Orazakhun”, “III-International”, “Sh-International” and “Dostyk and Co” and “Zhusaly”), 322 specimens were collected and measured as juvenile barbel. The percentage of barbel, according to the authors’ calculations, in the total number of recorded fish was about 0.59%. The size of juvenile barbel ranged with average values of 10.7 cm and weight of about 16.3 g, respectively [32].

In the spring of 2005, at the mouth of the river Syrdarya, staff of the Aral branch of the Kazakh Research Institute of Fisheries (Fisheries Research and Production Center) caught and examined 17 adult barbel individuals, which according to morphological characteristics were Aral barbels. Their standard length averaged 44.3 cm, and their weight was 1236 g. The numbers of females among the collected individuals were twice that of the males and their years ranged from 3 to 7 years of age. In the head part of the irrigation canals of the Kyzylorda hydroelectric complex in 2006, several specimens of barbel were found. When examining the irrigation canals of rice-growing farms in the Karmakshy district near the village of Zhusaly, it was possible to find barbels. It was assumed that juveniles of barbel in the irrigation system can be represented only by underyearlings, which probably grew from eggs that passively rolled through water intakes; however, two-year-old and three-year-olds were represented in the collections of different years. This may be due to the fact that not only eggs, but also young fish get into the irrigation system, as well as the fact that some of the fish caught in rice paddies can overwinter in collectors if they do not have time to freeze in winter, or in pits along the canal route, one of which served as a collect site for relatively large barbel specimens in 2006. At the same time, among the studied fish, there were individuals with morphopathological changes in the organs. The authors suggest that this may be the result of mixing in the irrigation system of individuals from relatively safe and heavily polluted areas of the basin, or the resistance of individual specimens to the effects of adverse factors [33].

According to information in Fishes of Kazakhstan (Mitrofanov V.P et al.) barbels reach sexual maturity at 5–8 years. Mass maturation of males occurs at the age of 6–7, females 7–8 years. Upon reaching puberty, after fattening in the sea, the Aral barbel made spawning migrations, rising along the rivers. Spawning aggregations in the sea before migration to the rivers began at a water temperature of about 16 °C. In March to early April, migrations can be observed in the river mouth, well-marked migrations of barbel to the rivers were usually noted in June–August. Barbels hibernated in the rivers, and in spring, with the onset of spawning temperature. Most of females, after reaching maturity, spawn in a season, which is first determined by the rate of accumulation of the necessary trophic and energy substances. The barbel spawns in riverbeds in areas behind sandy spits with a hard bottom, at a depth of 1–2.5 m and a well-defined, but not fast current. In the Syrdarya, the barbel begins to spawn in late April early May at a water temperature of 17–18 °C. Spawning continues all summer, and most intensively takes place at a temperature of 20–23 °C. Barbels’ spawning takes place at night. Sexual products are swept out in the water column, where fertilization takes place [4].

Barbel fishing in the Aral Sea was based on the fishers in age groups above five years to eight years and represented about 80% of the total harvest [35]. Maximum age of the Aral barbel reported from Kazakhstan was 22 years old and was caught in Syrdarya River near the city of Kyzylorda in the year 1959 [36]. Fishing of barbel in the 1950s was equal to roach, which occupied one of the first places in the Aral Sea in terms of fishing volume. Barbel was caught in the Aral Sea all year round, but about 90% of the total fishing was reported during the periods of mass spawning migration to river deltas [35].

There is no information about the biology of the Bulatmai barbel from the Aral Sea in the literature. Almost all available publications have been devoted to the barbel of rivers and reservoirs in other places. However in Kazakhstan, there are some literature on the barbel populations in the Shardara reservoir [37,38], Bogenskoe [39], and the rivers Bogen [40] and Chu [21]. The timing of reaching sexual maturity in the Bulatmai barbel is greatly extended. Age at first spawning of the Bulatmai barbel in different reservoirs ranged above 2 to 7 years. Females usually mature a year later than males and subsequently spawn with intervals of between 2–3 years [21,40,41,42]. Information about the nature and places of spawning of the Bulatmai barbel is contradictory. Some authors [43,44,45] believe that this fish spawns in rivers on sandy-stony ground; others [46,47] consider that it spawns in Central Asian reservoirs. Unfortunately, observations of the spawning and development of barbel eggs and larvae have not been described in the literature. The fecundity of the Bulatmai barbel is much lower than that of the Aral [48]. The maximum age of the Bulatmai barbel in the Bogen reservoir was reported as more than 10 years by Kuznetsova [49].

The Arys River flows on the territory of the South Kazakhstan region and is the main tributary of the river Syrdarya. A significant volume of the runoff in the middle reaches is used to irrigate agricultural fields through the Arys–Turkestan canal. In the lower reaches of the Arys River, the Shaulder hydroelectric power station is regulated; this leaves a certain imprint on the possibility of the ichthyofauna inhabiting the river. The migration, mainly of juvenile fish in the early stages of development, is possible throughout the river, and spawning and feeding migrations are possible only in limited areas. The main species here are of ichthyofauna is the Bulatmai barbel [5] and common marinka *Schizothorax intermedius* Mc’Clelland. High concentrations of the Bulatmai barbel are noted in the area where the Badam River flows in and downstream at a distance of up to 15 km. Basically, the population of the Bulatmai barbel is represented by individuals 25–40 cm long and weighing from 300 to 1200 g. The concentration of the barbel is from 8 to 31 fish for one kilometer of the Arys River at the confluence of the river Badam. The most diverse species composition of the ichthyofauna is in the lower section of the river Arys below the Shaulder hydroelectric power station. Due to the fact that the riverbed before confluence with the river Syrdarya is not regulated, there are both Aral and Bulatmai barbel, although the number of Bulatmai barbel here is lower than that of the Aral [5].

During a field expedition in the summer of 2011 [50], according to a report of the Fisheries Research and Production Center there were 30 Aral barbel specimens measured from among the rescued juveniles of Baubek-Baba LLP from rice paddies of the Karmakshy district. Their length without a caudal fin averaged 11.2 cm; body weight averaged 34.4 g, and 16 individuals of the Aral barbel were caught and the rest were presumably Bulatmai barbels. The length and weight of the caught fish were measured, the scales were taken to determine the age, and then they were released into the Syrdarya River. The ichthyological length of the fish fluctuated on average 38.06 cm. The weight of the barbels fluctuated on average 698 g, analyzing the size–weight indicators and the age composition of the barbels caught in the river. The Syrdarya is below the Aitek dam, it should be noted that these individual barbels constantly live in the Syrdarya River and breed there. The foregoing suggests that small self-reproducing populations of the Aral and Bulatmai barbels have survived in the lower reaches of the Syrdarya.

## 2. Results

### Recommendations for Conservation and Artificial Reproduction

The data obtained make it possible to recommend the capture of barbels from the Aral–Syrdarya basin and irrigation systems of rice farms to start work on the formation of artificial schools of these species and the development of biotechnology for their artificial breeding. Such work has already been carried out in past years [51] and has yielded encouraging results.

Carrying out these works is especially important for the purposeful formation of stocks of valuable fish species in the waters of the Small Aral Sea. Therefore, in 1992, a sand dam was built around Aralsk, which was supposed to protect the Small Aral from further drying out. A natural rise in water in early 1993 eroded it, and a new dam was created in 1997. It was destroyed in 1999 when the flow of the Syrdarya was increased. Instead, by 2005, the Kokaral dam was built at the expense of the World Bank, thanks to which the absolute water level in the Small Aral rose to 42 m [52].

Until now, we have only talked about the conservation of the existing extremely small populations of endangered species. However, what can be done to increase their numbers and thereby reduce the risk of complete extinction in the wild? A practical solution to this problem is possible only by removing the minimum required number of juveniles and spawners of rare species from their natural habitat, followed by artificial reproduction in fish farms of the Republic, for example, similar work is being carried out with the endemic Balkhash marinka [53].

The Aral and Bulatmai barbel are also included in the List of environmental protection objects of special ecological, scientific and cultural significance. Therefore, given that the preservation of the gene pool of such a valuable species as the Aral barbel is an exceptional need, it is urgent to take a set of measures to preserve and increase its numbers. In this case, a practical solution to the problem at hand is possible only with the removal of the minimum required number of barbel from the natural habitat, followed by artificial reproduction in the fish farms of the republic.

Research on the biotechnology of barbel breeding with subsequent acclimatization and reacclimatization of this species of fish will not only improve the composition of the ichthyofauna of the Aral Sea water bodies, but also preserve the genetic potential of natural populations.

For example, in China, *L. brachycephalus* has great potential for aquaculture. At present, *L. brachycephalus* is becoming the most economically significant due to its taste and high commercial value, and it has been cultivated in more than 20 provinces in China with an annual production of up to 20,000–40,000 tons. However, the genetic resources of *L. brachycephalus* are relatively limited, and genetic diversity is generally low due to inbreeding in small populations. Several studies have been conducted on artificial breeding, biological characterization and rearing of larvae [54,55]. Additionally, the authors of [56] noted the tolerance of the barbel to the salt–alkaline environment salinity < 10 g/L, alkalinity < 30 mmol/L.

To determine how many individuals should be selected to create replacement broodstocks, it is important to determine the effective population size to ensure the necessary genetic diversity [57]. How many individuals are needed to maintain the genetic diversity of a population? Franklin, I.R., The authors of [58] showed that 50 individuals can be considered the minimum number required to maintain genetic diversity. This figure is based on the practical experience of animal breeders, which shows that the group of selected animals should be increased after the loss of 2–3% of variability per generation.

However, many individuals in a real population do not reproduce due to age, poor health, infertility, emaciation, small body size, or community relationships that prevent some animals from mating. As a result of these factors, the effective size of the population of individuals participating in reproduction is significantly less than the actual size of the population. Since the rate of loss of genetic diversity depends on the effective size of the population, the loss of genetic diversity can be more rapid than that which can be expected from the size of the actual population [58].

For the reintroduction of the species in the lower section of the Syrdarya River, self-reproducing populations of the Aral and Bulatmai barbels have been preserved. Barbel individuals of different ages caught in sections of the river can serve as material for creating a replacement stock of barbel with subsequent offspring and stocking with fish. In addition, it is possible to collect juveniles from rice paddies with the subsequent rearing of the brood stock in a fish hatchery.

Unfortunately, none of the fish hatcheries registered in Kazakhstan specializes and is focused on the reproduction of juveniles of rare and endangered fish species. Their main direction is the cultivation of carp *Cyprinus carpio* Linnaeus, 1758, grass carp *Ctenopharyngodon idella* Valenciennes, 1844, silver carp *Hypophthalmichthys molitrix* Valenciennes, 1844. Thus, an appropriate infrastructure has not been created, a comprehensive action plan and specific implementation mechanisms for the implementation of laws and regulations have not been developed.

A retrospective analysis of the acclimatization, distribution and commercial value of the Aral barbel has been carried out. The limiting factors of its habitat and distribution have been determined.

It must be taken into account that when applying biotechnology, the genetic structure of replenishment from artificial reproduction must be identical to the natural genetic structure of populations. Thus, it is necessary to maintain the genetic diversity of artificial populations in accordance with the natural structure and the need to form a broodstock with a genetic structure similar to the spawning part of natural populations.

To determine the number of individuals required for removal from the natural environment, in order to form a broodstock, methodological approaches of research work were used. According to which the volumes of stocking of natural reservoirs with juveniles should correspond to the ecological receiving capacity of the reservoir.

The proposed place for the removal of barbel for scientific fishing, for further formation of the broodstock and the start of work on its artificial reproduction are the water bodies of the Aral–Syrdarya basin: the Syrdarya River from the Shardara reservoir to the confluence with the Small Aral Sea, directly from the Small Aral Sea (Figure 2), the Syrdarya River from downstream of the Shardara reservoir to the confluence with the Small Aral Sea.

According to our field sampling in the autumn period of 2021, according to the conducted research in rice paddies of the Kyzylorda region, during the reclamation work to save juveniles, we recorded 100 specimens of juvenile barbels (Figure 3).

Marked individuals after the removal of intravital parameters were released alive back into the water. Activities for collecting Aral barbel of different ages were carried out on the lower section of the Basykara dam (Kazalinsky district) of the Syrdarya River, and today work is underway on the site in the area of the Aitek dam, Zhalagashsky district.

For molecular genetic analysis, a small fragment (2 × 2 mm) of the pectoral fin was cut out from each individual using sterilized scissors. Tissues were immediately preserved in 96% ethanol.

For DNA extraction we applied the method from the previous work [53]. The amplification of cytochrome oxidase subunit 1 gene fragments was performed using the PCR technique. The primers used in the PCR process were FishF2_t1 TGT AAA ACG ACG GCC AGT CGA CTA ATC ATA AAG ATA TCG GCA C and FishR2_t1 CAG GAA ACA GCT ATG ACA CTT CAG GGT GAC CGA AGA ATC AGA A [59].

DNA barcoding has been proven to be a fast and accurate tool for a standardized molecular identification system [60,61].

After the sequence assembly and consensus editing, sequence reads were an average length of 585 base pair long. The database revealed maximum identity matches of 99% COI sequences belonging to *Luciobarbus* genera. For *Luciobarbus capito* there was more overlap in the sequences with Genbank Accession No. KM590440.1. Sequences obtained were compared to the data on GenBank for its identification, *Luciobarbus brachycephalus* had more matches with Sequence ID KP712068.1.

As a result of the study of the spring period of 2022, 25 specimens of barbels of different ages were caught, which are in cage and basin conditions to form replacement stocks using the domestication method.

Barbels of different ages were caught with fixed and rafting nets from different parts of the lower reaches of the Syrdarya River. The selected individuals of the barbel were placed in a plastic barrel with water with a capacity of 50 L, tightly closed with a lid on top, while the barrels were filled to the top with water. Up to three individuals of live barbel with a total weight of up to 5 kg were placed in one such barrel at a time. During transportation, the barrel must be in an upright position. The optimal time for transporting barbels from nets to the shore, to temporarily installed cages in a live-fish tank is up to 20–30 min.

In the spring, barbel individuals of different ages were transported from a natural reservoir to the Kamyshlybash fish hatchery in a live-fish vehicle with a capacity of 3 m^3^ with an oxygen supply (15 individuals), as well as to the Zhambyl fish farm (10 individuals). Transportation lasted from 1.5 to 2.0 h. Based on the work carried out, the survival rate of barbel spawners was determined when caught at the fishing grounds and transported to the fishery at up to 100%.

In a fish farm, barbel individuals were planted in pools of a recirculating water supply facility located in the incubation manufactory (Figure 4). Four basins were used for the domestication of barbels.

The formation of a replacement stock of valuable fish species by domestication is very promising, because it is possible to achieve a positive result already in a short period of time after maturation of spawners in artificial conditions. However, this method has a significant drawback, since the history of each individual is unknown, which makes it difficult to conduct selection and breeding activities. It should also be noted that barbels from natural water bodies are hardly accustomed to artificial food and often refuse to consume it, so the optimization of the fish feeding process occupies a key place in the biotechnology of domestication, since not all individuals adapt to artificial conditions.

When carrying out work on the domestication of fish, three adaptive fish-breeding tanks were prepared for pre-spawn keeping of barbel spawners, with an area of 3 m^3^. They were disinfected with a solution of potassium permanganate. All biological indicators were taken by the in vivo method (Table 1).

In hatcheries, there are two main ways to form broodstocks: cultivation of producers to a sexually mature state in artificial conditions; “from eggs”, domestication of sexually mature individuals caught in natural reservoirs (domestication).

The first method is based on the selection of the best offspring of barbel from planting material according to established criteria, followed by rearing of producers to a sexually mature state. This method is more labor intensive and time consuming. It provides for the selection of the best replacement individuals throughout the growing period by evaluating the conformation and fish breeding indicators. The undoubted advantage of this method is that all fish are well adapted to the conditions of detention, artificial feeding. The disadvantages include a high probability of inbreeding due to the limited initial producers and a long period of maintenance before first obtaining sexual products of 6–8 years. Despite the complexity of this method, in which great efforts are expended on the care, protection, maintenance and feeding of the repair, the positive point is the possibility of preserving the natural genetic diversity.

Therefore, at the first stage of the formation of a replacement broodstock, we can use only the second method, which involves the domestication of wild producers with subsequent maintenance in artificial conditions for the purpose of multiple intravital generations of offspring. The domestication of wild producers consists of obtaining reproductive products from them with further adaptation of fish to artificial conditions of detention and subsequent maturation. In domestication, mature producers are used. This method makes it possible to reduce the time of formation of the replacement broodstock by two to three times and to ensure sufficient heterogeneity of the formed school. The period of formation of the broodstock is reduced to a minimum—5–7 years. When forming the broodstock according to this principle, a selection method is used, which means assessing the physiological state of breeders and assessing their genetic characteristics to make it possible to select pairs of breeders more accurately and purposefully in order to obtain more viable offspring.

When laying the breeding material of barbel, it is necessary to take measures to ensure the preservation of the original genetic diversity. For this, larvae for breeding purposes must be selected from at least 10 (preferably 20) females. To fertilize the eggs of each female, the sperm of at least three males is used; eggs with a high percentage of fertilization (at least 80%) and a minimum number of embryos with deviations (2–3%) are selected. If these rules are observed, the increase in the level of inbreeding per generation will be minimal.

Thus, to ensure the optimal level of genetic diversity in conditions of artificial reproduction, it is necessary:–to maintain a certain level of the number of producers in the broodstock in the amount of: minimum 50, optimum 200;–during the reproduction of each generation, to ensure an equal contribution of representatives of different sexes to the spawning structure of the herd (the ratio of males and females is not more than 1:1), which will equalize the genetic contribution of each individual to the next generation,–to ensure that ecological and genetic monitoring of breeding material is carried out and, if necessary, introduce fish from natural populations.

In this connection, it is proposed to remove from the rivers from the natural environment within 3 years in Syrdarya, 200 barbel specimens of different ages (on average, weighing 1.5 kg with a total weight of 300 kg) for the formation of a replacement broodstock.

The data of the biotechnical breeding standards for the Aral barbel, developed in 1964 [52], and the optimal values, taking into account our attempts at multiple repetition, are presented in Table 2.

Taking producers is also supposed to be carried out on the section of the Syrdarya river from the Kazaly hydroelectric complex to the Shardara one. According to archive studies, the largest number of Aral barbel producers is found in these places. Selected barbel individuals of different ages in order to form a replacement broodstock is also possible in other parts of the river Syrdarya.

Harvested producers are planned to be concentrated in special coastal cages installed in the Syrdarya River. After that, some of the spawners are supposed to be transported for keeping in the pre-injection ponds of the Tastak site of the Kamyshlybash fish hatchery. Transportation is planned to be carried out in live-fish vehicles, or in live-fish vessels along the Syrdarya River (from the Kazaly hydroelectric complex to the Tastak site). Some producers plan to obtain sexual products directly at the aging sites. Producers are expected to be kept for up to 1 year. Incubation of eggs and stimulation of the maturation of gametes is supposed to be carried out using the pituitary gland of the carp. Ovulated eggs are more likely to be obtained a day after injection. The incubation of eggs obtained from spawners aged in cages in the current location in the Syrdarya River is planned to be partially carried out in the floating “Ses-Green” incubators installed in the current at a speed of 0.3 m/s. Eggs obtained from spawners aged in the pre-injection ponds of the Kamyshlybash fish hatchery, as well as the other part harvested at the spawners’ holding sites on the Syrdarya River, is planned to be incubated in the “Amur” apparatus located in the hatchery’s incubation manufactory. The choice of incubation apparatus is due to the positive buoyancy of the Aral barbel eggs.

At the end of the incubation of eggs, carried out under various conditions, keeping and growing all the larvae obtained under artificial conditions is planned to be carried out in trays located in the incubation manufactory of the Kamyshlybash fish hatchery. According to researchers, fry of the Aral barbel are omnivores, they prefer vegetation [7,23]. So, after reaching the required average weight (1.2–1.5 g), it is planned to grow juveniles in industrial conditions also for the formation of brood stocks.

Given the similarity in the biology of barbels, it is appropriate to apply these biotechnical recommendations to the Bulatmai barbel, actually this will be in our future research, which will still continue.

## 3. Conclusions

According to the International Union for Conservation of Nature (IUCN) freshwater ecosystems are among the most vulnerable [62,63]. Therefore, measures for the reintroduction of rare and endangered fish species into the water bodies which they originally inhabited, in order to restore their numbers are priority measures for the conservation of biodiversity, which, in turn, increases the productivity of the ecosystem and ensures its natural stability [62].

It is necessary to minimize the impact of various limiting factors, abiotic, anthropogenic, and biotic, while looking for alternative methods of conservation, such as the creation of specially protected natural areas, expanded artificial reproduction, and the development of commercial fish farming, which are currently poorly implemented.

The large-scale exploitation of natural resources over the past century has changed some of our country’s landscapes beyond recognition. In recent years, Kazakhstan has entered a period of rapid economic development, which will definitely lead to an increase in the anthropogenic pressure on natural resources. Subsequently, if adequate measures are not taken to reduce the anthropogenic impact on natural resources, including the ichthyofauna, this will lead to a reduction in the biodiversity of the animal world, and will primarily affect valuable fish species. Therefore, the barbel was included in the Red Book of the Republic of Kazakhstan in 2008.

The main document regulating the activities of the enterprises of the reproduction complex of Kazakhstan is the Law on the Protection, Reproduction and Use of the Wildlife of July 9, 2004 No. 593 [64]. This law provides for measures for the artificial breeding of objects of the animal world, including the use of their removal from their natural habitat with the subsequent formation of broodstock for the purposes of artificial breeding, conducting scientific research to develop methodological foundations for the reproduction of wildlife objects, which allows field research on the state of barbel stocks in the water bodies of the Aral–Syrdarya basin to be carried out and to assess a rare endangered fish species as barbels, for the formation of a replacement brood stock for artificial reproduction.

During the collection of results for the research conducted by the authors and the work on collected individuals of different ages, in the lower reaches of the Syrdarya River within the Kyzylorda region, select individuals, after recording intravital parameters, were planted in cage and basin conditions for the formation of replacement stocks by the method of domestication, and fins were also taken for molecular genetic analysis.

Currently, the only way to restore the amount of barbel is artificial reproduction, in order to preserve biodiversity and restore stocks of this valuable species in the natural environment and aquaculture. Barbels of different ages caught in the sections of the Syrdarya River will serve as material for the created repair-brood stock barbel with subsequent offspring. Under the current situation, the way out of this situation is seen in the creation of domesticated barbel breeding schools. In addition, it is possible to use collected juveniles from rice checks with subsequent rearing.

It should be noted that separate water basins of Kazakhstan are characterized by significant biodiversity. Under the conditions of increasing anthropogenic impact on the ichthyofauna of water basins, it becomes important to preserve their gene pool. This is especially for genuine species that are endangered. Recently, much attention has been paid in Kazakhstan to the conservation and restoration of valuable and endangered fish species.

When writing the review, archival and stock materials of the laboratories of the Fisheries Research and Production Center were used. To preserve these rare and valuable endemic species, individual genetic certification is required and their further careful protection by the state from unwanted illegal fishing.

The materials of this review can be used by nature conservationists, fisheries, and other organizations whose activities are aimed at the protection, reproduction and rational use of natural resources in order to create repair brood stocks for further conservation of biodiversity and restoration of its reserves in the natural habitat.

Research on the biotechnology of barbel breeding with subsequent acclimatization and reacclimatization of this species of fish will not only improve the composition of the ichthyofauna of the Aral Sea water bodies, but also preserve the genetic potential of natural populations. However, deficiency of information on genetic analysis of these endangered species may hamper efforts to protect, assess and restore populations.

## Figures and Tables

**Figure 1 biology-12-00489-f001:**
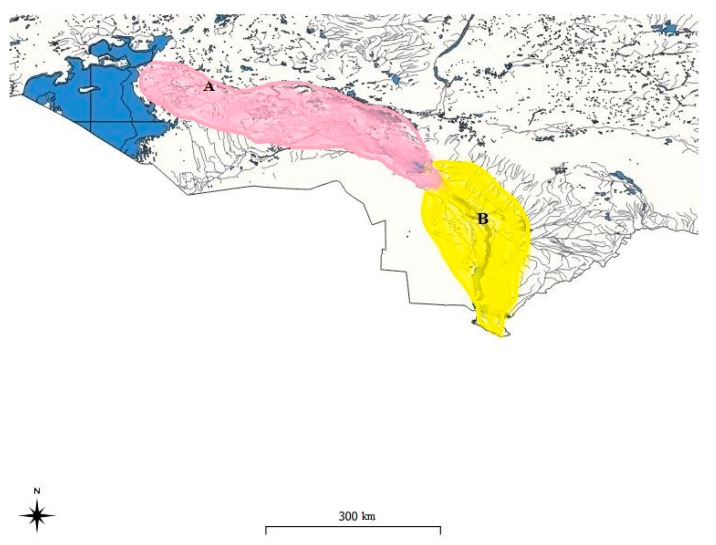
Distribution of the Aral barbel *Luciobarbus brachycephalus* Kessler, 1872 (**A**) and Bulatmai barbel *Luciobarbus capito* Güldenstädt, 1773 (**B**) in Aral—Syrdarya basin (Kazakhstan).

**Figure 2 biology-12-00489-f002:**
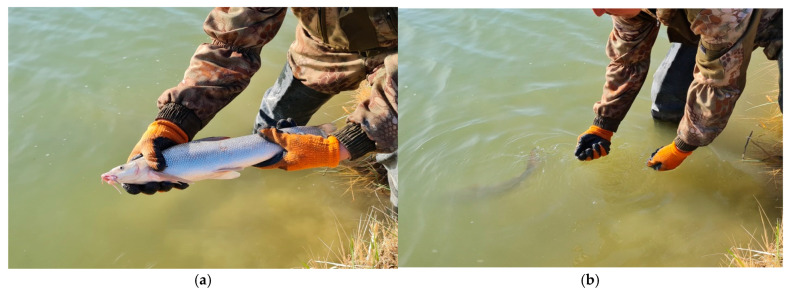
Collecting Aral barbel in the retaining zone of the Aral (Small) Sea: (**a**) one of the individuals caught in Small Aral Sea, 2021; (**b**) release of the barbel after biological measurements.

**Figure 3 biology-12-00489-f003:**
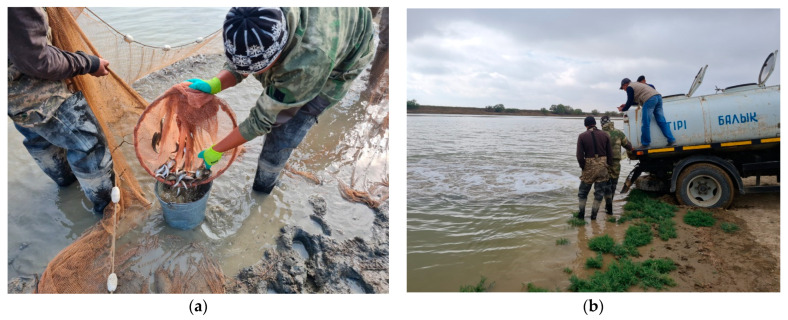
Juvenile barbels rescue in rice checks: (**a**) harvest of barbel from paddy checks; (**b**) transport of rescued barbel juveniles.

**Figure 4 biology-12-00489-f004:**
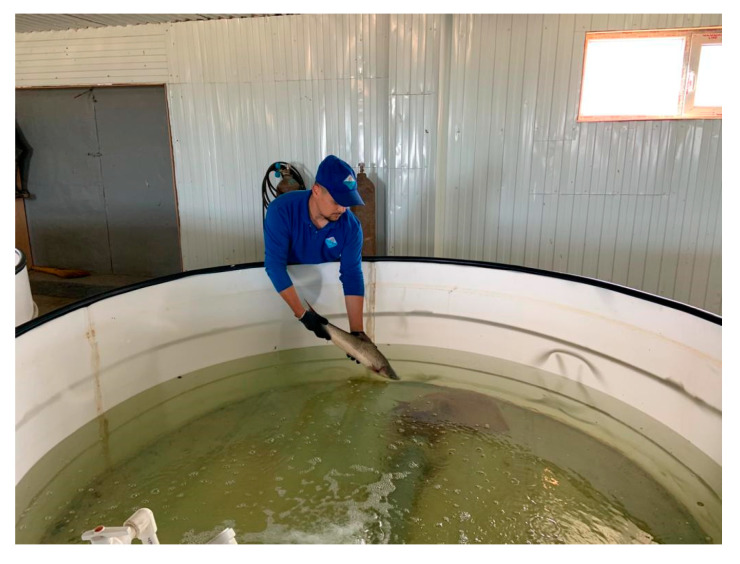
Seeding of selected Aral barbels for domestication in basin conditions.

**Table 1 biology-12-00489-t001:** Average size and weight characteristics of Aral barbel individuals of different ages.

Characteristics	Min	Max	M	*n*
L, mm	44.0	57.0	49.74	25
Q, gr	735.0	2140.0	1418.80	25

**Table 2 biology-12-00489-t002:** Biotechnical standards for artificial breeding of the Aral barbel.

Name	Units	Values	
		Standards 1964 y.	2022 y. (Present Work)
Working fertility	thousands of units	160–200	230–250
The amount of eggs in 1 g	thousands of units	360	360
Sex ratio	-	1:1	1:1
The waste of producers during the aging period:			
-long-term (up to 10 months)	%	30	30
-short-term	%	-	-
Dosage of the pituitary gland per 1 kg of fish weight	mg	2.0	2.3
Maturation of females upon injection	%	50	50
Losses of eggs during the incubation period	%	8	14
Losses of larvae for transportation at a stocking density of 5–6 thousand units per liter	%	5	5
Losses of larvae during overexposure up to 6 days	%	3	3
The norm of laying eggs on 1 Ses-Green apparatus (100 × 60 × 30 cm)	thousands of units	40 66	40 116–125
Optimal incubation temperature	°C	20	22
Keeping larvae on 1 m area	thousands of units	60	60
Stocking density of larvae in ponds	thousands of units	350–400	until 400
Losses of juveniles during the period of rearing in ponds	%	35	35
Output of juveniles from 1 ha of ponds	thousands of units	225–260	260
Average weight of juveniles	gr.	1.0	1.2
Growing time	months	1.5	2.5
Survival of underyearlings from the beginning of hatching of larvae	%	55	55
Fish productivity of ponds with the introduction of higher aquatic vegetation	kg/ha.	200	300

## Data Availability

The data presented in this study are available from the authors at adyrbekova@fishrpc.kz.

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
