# Peer review of "Current State of Rare and Endangered Barbels of the Genus Luciobarbus Heckel, 1843 in the Aral–Syrdarya Basin (Kazakhstan) and Prospects for Their Conservation (A Review)"

_biology, 2023, doi:10.3390/biology12040489_

Round 1

Reviewer 1 Report (New Reviewer)

Dear authors,

Really, the restoration of rare species often requires their artificial reproduction. Barbel are no exception, therefore the submitted manuscript is interesting. However, there are significant remarks for the manuscript.

1.   It is unlikely that this manuscript can be positioned as a Review, given that the work contains a lot of its own observed data: from own finds of barbels in different parts of river basins and their characteristics, to information about the biotechnology of domestication and new data on Biotechnical standards for artificial breeding of the barbel.

2.   It is necessary to more clearly structure the text: for example, about the reasons for the decrease of barbel in line 67, 105-107, 279; About the results of own finds first lines 127-152, then at 195-203. Data on biology of different species should be presented sequentially.

3.   The first figure needs clarification. The text says that "L. capito is also represented by two populations: in the southern part of the Caspian but not found in Kazakhstan and in the Aral Sea basin” (line 38), but this is not clear from the figure. It is necessary to bring the full  area of species, not limited of Kazakhstan. In addition, it will be useful to illustrate the distribution areas of individual populations of both barbel species, about which you write in the text.

4.   Line 67, about «resorption of fish oocytes, and the death of juveniles in irrigation systems» and Lines 105-107? about «The main factors that influenced the commercial stocks of barbel are: deterioration of the conditions of natural reproduction; intensive fishing; a large by-catch of juveniles in fixed nets; mass mortality of juveniles in irrigation systems; illegal fishing». This requires evidence: substantiating such an opinion or links to previously published data.

5.   Lines 154 and further. It is not clear, is this your own data? If your own, you need to indicate where and how they are collected, if literary - we need links to specific work. Or is it all simply rewritten from generalization Mitrofanov, V.P.; Dukravets, G.M.; Melnikov, V.A.; Baimbetov, A.A. Fishes of Kazakhstan, 1988?

6.   Lines 218-226; 232-251 is more likely a text for the introduction and justification of the relevance of the work, or may be in the discussion, but definitely not in “Results”.

7.   Why differences in Biotechnical Standards for Artificial Breeding of the Aral Barbel in 1964 and 2022? You should justify changes in the values of indicators. Have there been experiments?

8.   You give biotechnical standars for aral barbel. Will the standards for Bulatmai barbel differ?

9.   It is not necessary to describe in detail the history of specific The reproduction complexes, where barbel reproduction works are supposed to be carried out. Or do you think that this will not work in other fish farms?

In my opinion, the manuscript should be revised and significantly reduced in order to be accepted for publication.

Author Response

Reviewer 2 Report (New Reviewer)

This is a review describing the current state of rare and endangered barbels of the genus Luciobarbus(Luciobarbus brachycephalus Kessler, 1872 and Luciobarbus capito Güldenstädt, 1773)in the Aral-Syrdarya basin (Kazakhstan) and prospects for their conservation. This has important implications for the conservation of these two species. The presentation was well written with enough problem statements. However, there are specific comments are as below:

1. Line 124-126: “It should be …… 125 into wastewater.” Is there any data or literature to support it?

2. Line 137-138: “Several copies …… detailed study.” Have they formed a research report? If so, please give a brief introduction.

3. Line 166-167: “Maximum age of the Aral barbel …… in year 1959.” Please add the body length and weight of the fish if available.

4. For other species appearing in the text, such as roach, carp, grass carp and silver carp, please add Latin scientific names of them.

5. Line 180-181: “The maximum age of the bulatmai barbel …… 10 years by Kuznetsova.” Please add the body length and weight of the fish if available.

6. It is suggested that which kind of barbel is indicated in Figures 2-4?

7. Line 375-376: “The method of domestication of spawners …… in fish farms is more difficult.” It is suggest a brief description of which aspects?

8. Line “Stocking density of larvae in ponds” in Table 2, what is meaning of  Ð´Ð¾ 400” ?

9. Please check the “,” of “1,0”, “1,2”, “1,5” in table 2. “,” whether should be “.”?

10. Please check the citation format of the references. Many of the references were in inconsistent format.

Author Response

Reviewer 3 Report (New Reviewer)

This is a good, solid data paper about an important topic. The fish of the Aral Sea and its basin are in desperate shape because of near-drying of the sea and heavy use of the rivers for agriculture, hydropower, and urban areas.  This paper not only documents the problem for two species of barbels, but provides detailed and explicit instructions on how to alleviate and potentially fix the problem, at least for the Aral barbel. The difficulties of aquaculture under the circumstances, with a previously uncultivated fish species, are not underestimated. 

The English of the paper needs a good deal of work. Spelling and usage are often not standard. Aral should always be capitalized in the name Aral barbet, for one obvious thing. 

Line 219 should be clarified to show that the figure that one-third of species are endangered refers to freshwater fish, not all species on earth.

Otherwise, this is an interesting, important, well-written paper from an underreported area.

Round 2

Reviewer 1 Report (New Reviewer)

Dear Authors,

In my opinion, the manuscript has become much better.

I wish you success in your difficult and noble cause.

This manuscript is a resubmission of an earlier submission. The following is a list of the peer review reports and author responses from that submission.

Round 1

Reviewer 1 Report

Dear Authors

I had a chance to review  the manuscript entitled: “Current state of rare and endangered barbels of the genus Luciobarbus
Heckel, 1843 in the Aral-Syrdarya basin (Kazakhstan) and prospects for their
conservation (A Review)” 
Manuscript ID: biology-2123734 submitted to Journal Biology, which aims to review the status of the genus Luciobarbus in Kazakhstan by providing some information about the taxonomic status, distribution, and fisheries.  The ms can be obviously of great interest to the ichthyologists and fisheries biologists if presented in a good shape and format. In my opinion, this manuscript is of interest for publication in the Journal of biology, but not in the present form because it needs major consideration and revision.

The comments and corrections are all given in the attached file. Some of the main issues are listed here:

General common

1-    The authors have not discussed the current status of the genus Luciobarbus in Kazakhstan following recently published articles. Currently, no subspecies are considered for fish. I suggest the authors to follow the catalog of fishes (2022):

https://researcharchive.calacademy.org/research/ichthyology/catalog/fishcatmain.asp

 to discuss it.

2-    All the scientific names should be in italics and double-checked following catalog of fishes (https://researcharchive.calacademy.org/research/ichthyology/catalog/fishcatmain.asp).

3-    The morphological data have not been provided in detail.

4-    The molecular data have not been reviewed.

5-    A good map to show the distribution range of species is needed.

6-    The issues regarding the conservation of species have not been well provided and discussed.

7-    The statistical data should be presented in the form of clear tables or figures.

8-    The ms should have several distinct subsections related to each given topic.: Taxonomy, Morphology, Biology, Ecology, Fisheries, Conservation, and management,...

9-    Extensive editing of the English language and style is required.

Author Response

Dear reviewer, we really appreciate your valuable comments, all have been taken into account as much as possible.

Sincerely.

Reviewer 2 Report

Dear Authors,

Dear Authors,

It is very important to study on the conservation status of endangered species. It is also very important to communicate the unique issues in different localities, which will help to propose and formulate conservation measures in other parts of the world. Your study is a very interesting and which will be an eye-opener to others who are taking decisions related to natural aquatic resources management. However, I have observed the following weaknesses in your manuscript. I believe your manuscript can improve further and will be an important publication for others in future. Please see the following comments.

Authors must emphasize the research question and the objectives of the study. However this message is not clearly conveyed in this manuscript

Your literature review is not well organized. Information flow is not in a proper order.  Information should interpret in an organized manner to lead your reader to the exact issue you are going to address. Although lot of information are available, those are not presented in a well-organized manner.

There should be rigorous scrutinizing needed in the language and writing style. Please see the attached manuscript for some corrections and suggestions

It is important to indicate the protocol of your literature review in the manuscript.

Include a suitable map in the manuscript to indicate the important localities that are explained in this study.

Result section includes the methodology adapted in the research trials. Separate it in to methodology and results.

Author Response

(The authors gave the same response as above.)

Reviewer 3 Report

The manuscript seems to contain very important and valuable data on barbels endangerous species and has the potential to be a good contribution to the area.

Unfortunately, it presents many grammar issues, and before revising it, I suggest sending it to an English review. After an English review, I will be happy to receive it again to give my suggestions. 

Author Response

(The authors gave the same response as above.)
